# Antibacterial gene transfer across the tree of life

**Jason A Metcalf[1], Lisa J Funkhouser-Jones[1], Kristen Brileya[2], Anna-Louise Reysenbach[2], Seth R Bordenstein[1,3]\***

[1]Department of Biological Sciences, Vanderbilt University, Nashville, United States; [2]Department of Biology, Portland State University, Portland, United States; [3]Department of Pathology, Microbiology, and Immunology, Vanderbilt University, Nashville, United States

**Abstract** Though horizontal gene transfer (HGT) is widespread, genes and taxa experience biased rates of transferability. Curiously, independent transmission of homologous DNA to archaea, bacteria, eukaryotes, and viruses is extremely rare and often defies ecological and functional explanations. Here, we demonstrate that a bacterial lysozyme family integrated independently in all domains of life across diverse environments, generating the only glycosyl hydrolase 25 muramidases in plants and archaea. During coculture of a hydrothermal vent archaeon with a bacterial competitor, muramidase transcription is upregulated. Moreover, recombinant lysozyme exhibits broad-spectrum antibacterial action in a dose-dependent manner. Similar to bacterial transfer of antibiotic resistance genes, transfer of a potent antibacterial gene across the universal tree seemingly bestows a niche-transcending adaptation that trumps the barriers against parallel HGT to all domains. The discoveries also comprise the first characterization of an antibacterial gene in archaea and support the pursuit of antibiotics in this underexplored group.

**\*For correspondence:** s.bordenstein@vanderbilt.edu

**Competing interests:** The authors declare that no competing interests exist.

**Reviewing editor**: Wilfred van der Donk, University of Illinois-Urbana Champaign, United States

## Introduction

HGT is rampant among prokaryotes and phages and is an important mechanism for acquisition of new genes and functions (*Popa and Dagan, 2011*), including the shuttling of antibiotics and antibiotic resistance between bacteria (*Clardy et al., 2009*). Instances of interdomain horizontal transfer of diverse genes between two domains of life or between viruses and their hosts are also increasingly documented (*Nelson et al., 1999*; *Husnik et al., 2013*; *Dunning Hotopp et al., 2007*; *Wu et al., 2013*; *Gladyshev et al., 2008*; *Bratke and McLysaght, 2008*; *Danchin et al., 2010*). While a minority of these transfers have been functionally investigated, the biological activity, selective advantages, and ecological contexts of many interdomain HGT events remain poorly characterized (*Dunning Hotopp 2011*, *Keeling and Palmer, 2008*). In comparison to these intradomain or interdomain highways of HGT (*Beiko et al., 2005*), independent transmission of the same gene family to archaea, bacteria, eukaryotes, and viruses is extremely uncommon and subject to apparently rare events throughout the history of life (*Moran et al., 2012*; *Lundin et al., 2010*; *Koonin et al., 2003*; *McClure, 2001*; *McDonald et al., 2012*). When taken together, genome-enabled studies suggest that horizontal gene transfers (HGTs) are biased and experience a frequency gradient that decreases from within domain > between two domains > between all domains of life (*Bruto et al., 2013*; *Zhaxybayeva and Doolittle, 2011*; *Puigbo et al., 2009*; *Andam and Gogarten, 2013*, *2011*). However, the fact that a large number of organisms from all domains of life have been artificially transformed with genes evolved in other taxa indicates that there is no fundamental obstruction to interdomain HGT when barriers to transfer are deliberately removed.

One significant question then is why do highways of intra- or interdomain transfers occur more frequently than transfers to all domains in the universal tree of life? There are at least two explanations.

**eLife digest** Living things inherit most of their genetic material from their parents, so genes tend to be passed on from one generation to the next—from ancestors to descendants. Sometimes, however, DNA is transferred from one organism to another by other means. These events, collectively called horizontal gene transfer, are fairly common in nature; genes have been passed between different species as well as between different groups of organisms. For example, genes that confer resistance to antibacterial drugs have transferred from one species of bacteria to another, and other genes have also 'jumped' from bacteria to plants or animals.

Now Metcalf et al. have studied a gene that first arose in bacteria and that encodes an enzyme called a lysozyme. This enzyme breaks down the outer casing of a bacterial cell: a step that is required for a bacterium to reproduce and divide in two. When Metcalf et al. searched for relatives of the lysozyme gene, they found copies in many other species of bacteria and revealed that this gene has been repeatedly transferred between different bacteria. Members of the lysozyme gene family have also 'jumped out' of bacteria and into other organisms at least four times. Metcalf et al. found related lysozyme genes in a plant, an insect, many species of fungi, and a single-celled microbe (called an archaeon) that lives at hot, deep-sea vents.

A gene family being spread this widely across the tree of life has not been seen before. Nevertheless, as DNA is a common biological language to all living things, it is likely that all the different species that have received a lysozyme gene might use it for similar purposes.

Metcalf et al. reveal that the lysozyme could be being used as an antibacterial molecule. The archaeon lysozyme can kill a broad range of bacteria; and when the gene was transferred into *Escherichia coli* bacteria, only the bacteria that mutated the lysozyme gene to render it useless were able to survive. Metcalf et al. also revealed that the archaeon microbe produces more of the enzyme if bacteria are present, which allows it to outcompete these bacteria.

These findings suggest that there may be a number of horizontally transferred genes that have antibacterial activity against a wide range of bacteria. Searching for these genes—particularly in the largely underexplored group of archaea—might reveal new sources for antibiotic drugs to treat bacterial infections.

First, recurrent transfer of the same gene family may be limited by incompatible mechanics of gene transfer (e.g., transduction, transfection, plasmid exchange, isolation of eukaryotic genome in nucleus, separation of somatic and germline tissues in multicellular eukaryotes) between domains compared to within domains. However, the individual success of gene transfers between any two domains of life, for example archaea and bacteria (*Nelson et al., 1999*; *van Wolferen et al., 2013*), bacteria and eukaryote (*Andersson, 2005*; *Bordenstein, 2007*; *Gladyshev et al., 2008*; *Danchin et al., 2010*), and archaea and eukaryote (*Andersson et al., 2003*; *Schonknecht et al., 2013*), suggests that these barriers may be minimal. Second, the selective barriers against HGT of the same gene to multiple taxa and preservation of the gene through evolutionary time are multifaceted given the potential costs associated with HGT (*Baltrus, 2013*), and that each recipient may not benefit from the trait conferred. In the case of selfish genetic elements, HGT provides a strong benefit to the gene itself, yet these genes can be detrimental to the host organism and select for the evolution of countermeasures to gene propagation (*Werren, 2011*). Thus, it may be necessary for a transferred gene to confer a benefit to its new host in order to be stably maintained in the host genome over the long term. Given these evolutionary dynamics, there may be very few niche-transcending genes (*Wiedenbeck and Cohan, 2011*), defined as genes that are useful in different physiological capabilities, cellular structures, and ecological niches, which repeatedly increase fitness of each recipient across the whole diversity of life and can be stably and repeatedly transferred between very divergent organisms.

Among the few putative cases, there is a pore-forming toxin domain that appears to have been anciently transferred between diverse lineages (*Moran et al., 2012*). However, the distribution of the transfer across the tree of life is unclear because archaeal sequences were not included in phylogenetic analyses due to low support values. Other candidate genes encode proteins involved in nucleotide metabolism, intramembrane proteolysis, or membrane transport, but the transfer events defy clear interpretations due to their deep antiquity in evolutionary time and the confounding issues of ancient

paralogy (*Lundin et al., 2010*; *Koonin et al., 2003*; *McClure, 2001*; *McDonald et al., 2012*). Moreover, these transfers are often not functionally validated in the recipient taxa.

Here we demonstrate for the first time, to our knowledge, that a functional antibacterial gene family scattered across the tree of life in diverse ecological contexts. This bacterial gene encodes a glycosyl hydrolase 25 (GH25) muramidase, a peptidoglycan-degrading lysozyme that hydrolyzes the 1,4-β-linkages between N-acetylmuramic acid and N-acetyl-D-glucosamine in the bacterial cell wall. Typically found in bacteria (*Cantarel et al., 2009*), the lytic enzyme classically functions in cell division and cell wall remodeling (*Vollmer et al., 2008*), while in bacteriophages they lyse bacterial peptido-glycan at the end of the phage life cycle (*Fastrez, 1996*). Although members of the GH25 muramidase family have been noted in other taxa (*Korczynska et al., 2010*; *Nikoh et al., 2010*), extensive analysis of their evolutionary history and functions have not been undertaken. We hypothesized that the trans-fer of antibacterial genes from bacteria to archaea and to eukaryotes bestows a niche-transcending adaptation that overcomes the barriers against repeated and evolutionarily stable HGT of the same type of gene across the tree of life.

## Results and discussion

### A bacterial GH25 muramidase is present in all domains of life

During a homology search, we uncovered 75 nonredundant sequences (E-values $\leq 10^{-12}$) of a bacterial GH25 muramidase in disparate taxa across the tree of life, indicating possible HGT of a bacterial gene to both eukaryotic and archaeal species as well as to phages. Putative HGT events were identified in the genomes of the plant *Selaginella moellendorffii* (*Banks et al., 2011*), the deep-sea hydrothermal vent archaeon *Aciduliprofundum boonei* (*Reysenbach et al., 2006*), the pea aphid *Acyrthosiphon pisum* (*International Aphid Genomics Consortium 2010*, *Nikoh et al., 2010*), and several species of fungi such as *Aspergillus oryzae* (*Machida et al., 2005*). We verified the presence of the lysozyme gene in natural populations of selected HGT recipients by PCR and sequencing of the GH25 murami-dase domain (*Figure 1—figure supplement 1*), including *Aciduliprofundum* field samples harvested from hydrothermal vents worldwide. We detected lysozyme genes in 9 out of 12 field isolates of *Aciduliprofundum* from deep-sea vents in the Atlantic and Pacific oceans, 5 out of 6 species in the plant genus *Selaginella*, and 8 out of 9 aphid species in the subfamily Aphidinae (*Supplementary file 1*). However, it is possible that PCR-negative strains actually do possess lysozyme genes and were simply more divergent than could be detected with our primers. At the protein level, sequenced field isolate lysozymes were relatively similar to each other in each clade, with 74% pairwise identity (385aa alignment) amongst *Aciduliprofundum* sequences, 85.1% (203aa alignment) amongst three intact *Selaginella* sequences, and 87.3% identity (93aa alignment) amongst Aphidinae sequences, with more variable divergence between clades (full identity matrix available in *Supplementary file 2*). We also found lysozymes in two additional WO phages as part of an ongoing next generation sequencing project of *Wolbachia* viruses (unpublished data).

To rule out spurious bacterial contamination in these genomes and to confirm genomic integration of the lysozyme gene, we employed direct sequencing of PCR products amplified using primers inside the lysozyme gene paired with primers outside the gene for *Aciduliprofundum* field samples and *S. moellendorffii*. Incorporation of the lysozyme gene was verified in all cases tested (*Figure 1—figure supplement 2*). Additionally, *Aciduliprofundum* samples were grown in strict monocultures as deter-mined by 16S amplicon monitoring. Integration of the *A. pisum* lysozyme has been previously estab-lished (*Nikoh et al., 2010*). Flanking genes in the recipient genomes were non-bacterial on either side of the transferred lysozyme in each case (*Figure 1*, *Supplementary file 3*), with two excep-tions. A bordering gene in *A. boonei*, ADP-ribose-1″-monophosphatase (App-1), possesses both bac-terial and archaeal homologs and a phylogenetic analysis suggests HGT unrelated to the lysozyme transfer (*Figure 1—figure supplement 3A*). This transfer was likely between archaea and Thermotogae bacteria. The second exception is a GH2 hydrolase gene adjacent to the lysozyme in *A. oryzae*. This hydrolase has bacterial homologs in the phylum Actinobacteria, and recapitulates the same phylo-genetic pattern seen in the GH25 muramidase (see below, *Figure 1—figure supplement 3B*). Thus, it is likely that the lysozyme and GH2 hydrolase were transferred to fungi in a single event. GH2 hydrolases have a number of carbohydrate-degrading enzymatic abilities (*Cantarel et al., 2009*), including β-galactosidase, β-mannosidase, and β-glucuronidase activities, suggesting that this gene may benefit the fungi by adding additional digestive/nutritive capabilities in Dikarya.

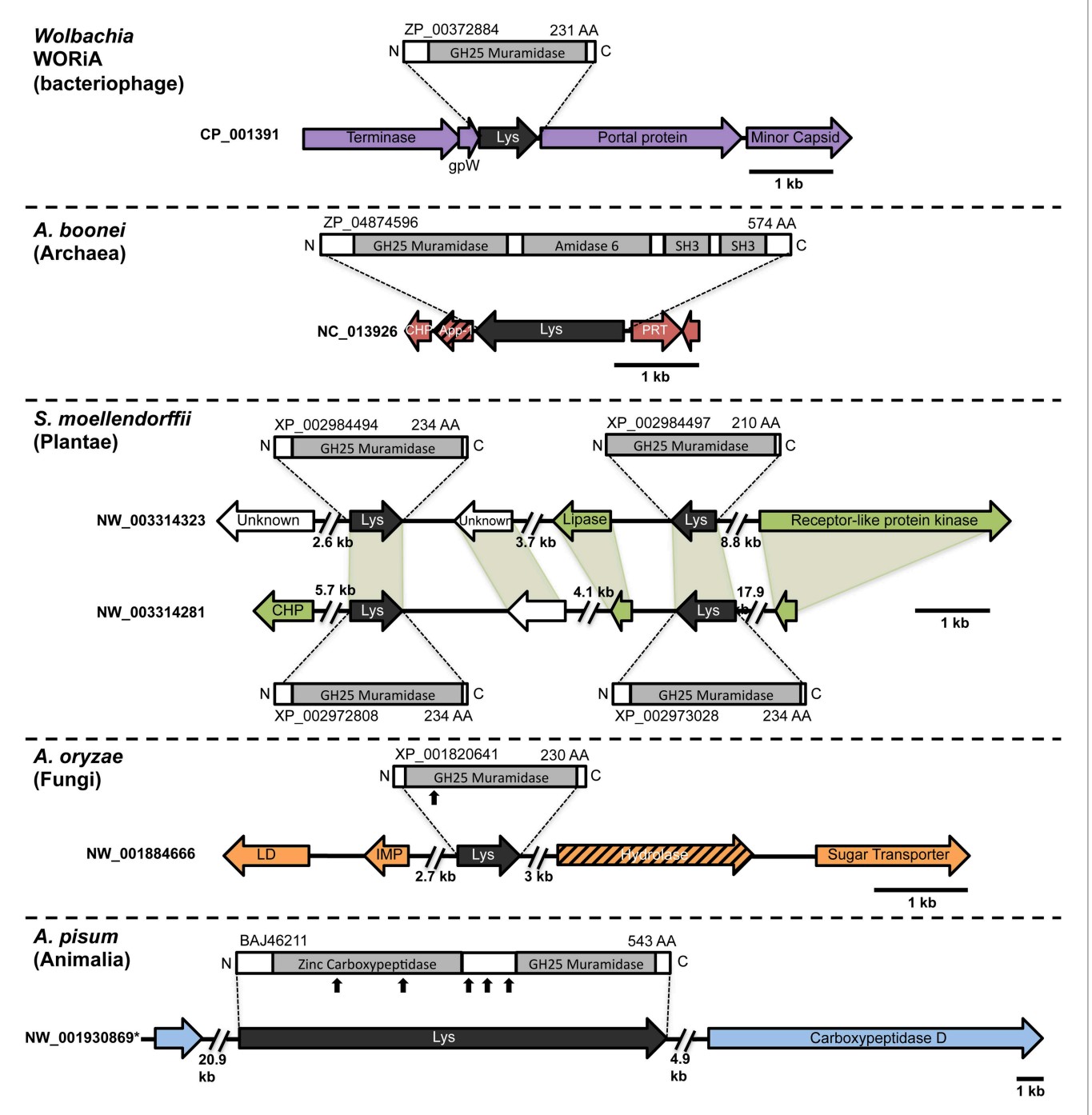

**Figure 1**. Architecture of HGT candidates and surrounding genes. Each arrow represents an open reading frame transcribed from either the plus strand (arrow pointing right) or the minus strand (arrow pointing left). The color of the arrow indicates the taxa the gene is found in based on its closest homologs. Black = Eubacteria, purple = virus, red = Archaea, green = Plantae, Orange = Fungi, Blue = Insecta, white = no known homologs, dashed line = present in multiple domains. The length of the arrows and intergenic regions are drawn to scale except where indicated with broken lines. The four paralogs of the lysozyme in *S. moellendorffii* occur on two genomic scaffolds with light green bands connecting homologous genes. Vertical arrows indicate the location of introns in the *A. oryzae* and *A. pisum* lysozymes. Abbreviations: Lys: lysozyme, gpW = phage baseplate assembly protein W, SH3: Src homology domain 3, App-1 = ADP-ribose-1"-monophosphatase, PRT = phosphoribosyltransferase, LD = leucoanthocyanidin dioxygenase; IMP = integral membrane protein. A protein diagram for each lysozyme is drawn to scale with the light gray regions highlighting a conserved protein domain. *A. pisum* diagram is based on Acyr_1.0 assembly and transcription data (**Nikoh et al., 2010**); the annotation in Acyr_2.0 is different.

*Figure 1. Continued on next page*

*Figure 1. Continued*

The following figure supplements are available for figure 1:

**Figure supplement 1**. Presence of HGT lysozyme genes in field samples.

**Figure supplement 2**. PCR amplifications testing genomic integration with primers within and outside of lysozyme genes.

**Figure supplement 3**. Protein phylogeny of neighboring genes to transferred lysozymes.

## Non-bacterial GH25 muramidases arose from HGT

To establish parallel HGT, i.e., the independent transfer of the same gene family to multiple lineages, we conducted a phylogenetic analysis on 86 GH25 muramidase sequences using Bayesian and maximum likelihood inference methods (*Figure 2A*). We combined non-redundant *Aciduliprofundum*, *Selaginella*, and WO sequences obtained from PCR and Sanger sequencing with blastp results to reconstruct the phylogeny. Putative instances of HGT are diagrammed in *Figure 3*. Additionally, transferred lysozymes in nonbacterial taxa were used as queries to identify homologs and make a second set of phylogenetic trees to confirm the HGT (*Figure 2B–D*, *Figure 2—figure supplements 1 and 2*, *Supplementary file 4*). Three key results emerge from these phylogenetic analyses: (i) at least three independent instances of interdomain HGT of the bacterial GH25 muramidase occurred in nonbacterial taxa (*Aciduliprofundum*, *Selaginella*, and *Insecta*) as well as a number of transfers to bacteriophages, (ii) vertical transmission of the transferred gene ensues in some descendant taxa (i.e., *Aciduliprofundum* and *Selaginella*), and (iii) frequent HGT of the muramidase between bacterial clades accompanies the interdomain transfer, indicating unusually frequent and broad-ranging HGT of this niche-transcending gene family.

To statistically validate parallel HGT across the tree of life, we performed a Shimodaira–Hasegawa test (SH-test) (*Shimodaira and Hasegawa, 1999*) by comparing our consensus tree (*Figure 2A*) against a hypothetical tree with a binary constraint in which bacterial sequences are monophyletic and separate from monophyletic nonbacterial sequences. In this hypothetical tree consistent with the tree of life, lineage relationships within the bacterial and nonbacterial groups were permissively set as unconstrained. Results indicate that the hypothetical tree is significantly worse than the HGT tree, as expected (p<0.01, D(LH) = −133.9, SD = 31.5). We repeated this analysis with the hypothetical tree compared to 100 randomly sampled HGT trees from maximum likelihood bootstrapping and found the hypothetical tree was also worse than each of these trees (p<0.01). Finally, we performed SH tests between the HGT tree and either 1) a three-domain constraint tree or 2) a monophyletic eukaryote branch constraint tree, and again found that the constraint trees were significantly worse than the HGT tree (p<0.01). Thus, the null hypothesis of vertical descent is rejected, even under the most permissive conditions.

We observed that each interdomain HGT event (*Figure 2*) occurred between taxa that coexist in the same ecological niche, a likely prerequisite for HGT. For instance, the *A. boonei* lysozyme is in a clade dominated by Firmicutes whose members can be common in deep ocean sediments (*Orcutt et al., 2011a*). Indeed, *Bacillus* species have even been found in hydrothermal vents of the same fields in which *Aciduliprofundum* strains were isolated (*Reysenbach et al., 2000*). The *A. pisum* lysozyme clade includes *Wolbachia* prophages and Proteobacteria, which are common endosymbionts of aphids and other insects (*Gomez-Valero et al., 2004*; *Augustinos et al., 2011*; *Wang et al., 2014*). The *S. moellendorffii* plant lysozyme is closely related to Actinobacteria, which are dominant microbes in soil (*Bulgarelli et al., 2013*). These associations, while not proof of HGT, establish interactions that may have facilitated the transfers, although any number of intermediate gene carriers is possible. While the phylogenetic pattern of the GH25 muramidase found in fungi is consistent with HGT (*Figure 2A*, *Figure 2—figure supplement 2*), the transfer occurred anciently in fungal evolution prior to the divergence of Dikarya, as the domain is present in both Basidiomycota and Ascomycota, but not other fungal phyla. As is the case with most putative ancient transfers, the deep branches of the tree are poorly supported and a definitive donor taxon cannot be established. Additionally, the possibility of multiple ancient transfers between bacteria and fungi or among fungi and plants cannot be excluded. However, a nucleotide-level phylogeny also supports HGT from an ancestral Actinobacterium (*Figure 2—figure supplement 3*).

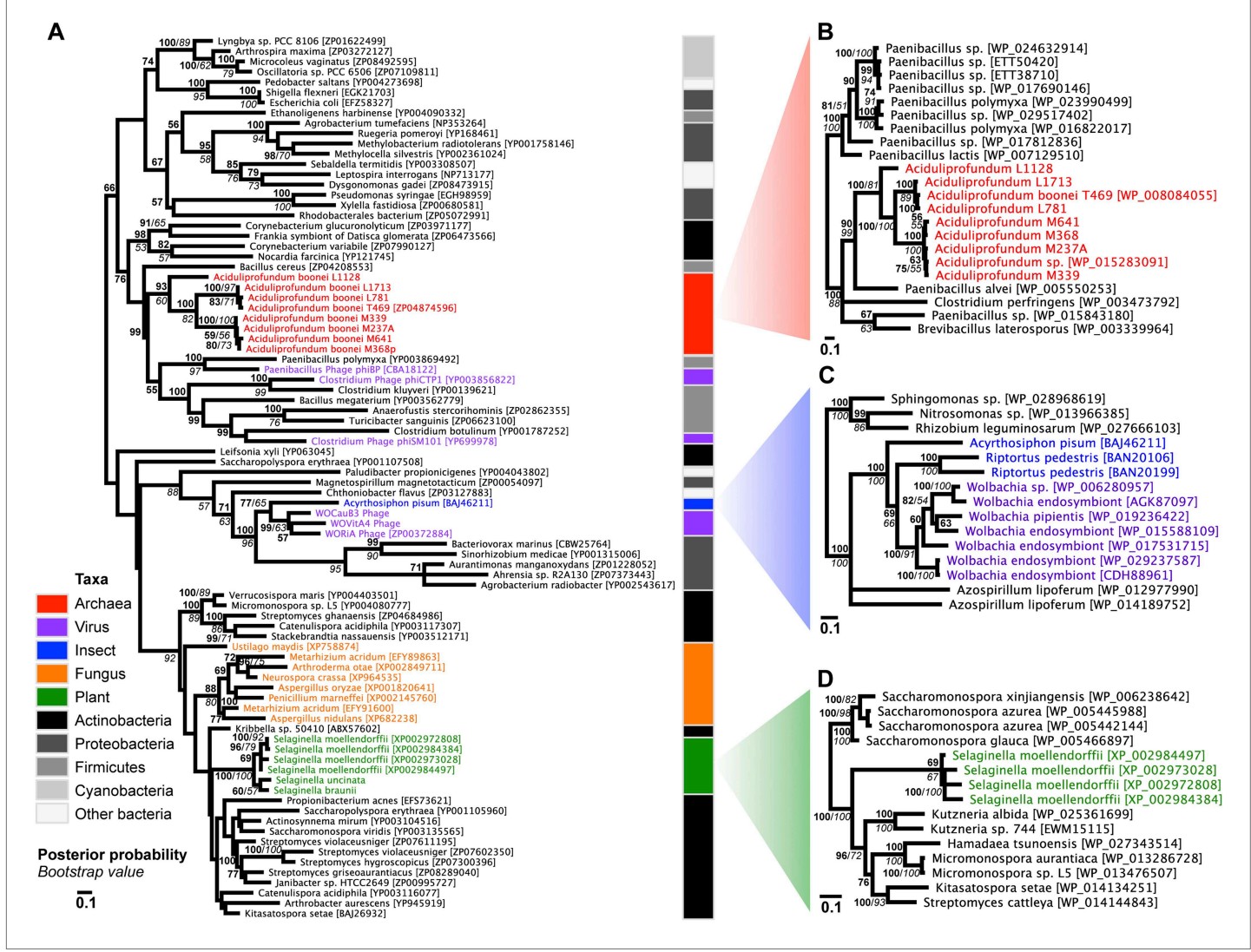

**Figure 2**. Phylogeny of GH25 muramidase. (**A**) Phylogeny based on alignment of 113aa without indels consisting of top E-value hits to blastp using WORiA phage lysozyme as a query. Taxon of origin for each amino acid sequence is indicated by color. Posterior probability (Bayesian phylogeny) and bootstrap values (maximum likelihood phylogeny) are indicated at all nodes with values above 50. Branch lengths represent number of substitutions per site as indicated by scale bar. Tree is arbitrarily rooted. Iterative phylogenies based on top E-value blastp hits to *A. boonei* lysozyme (**B**), *A. pisum* lysozyme (**C**), and *S. moellendorffii* lysozyme (**D**) are also shown.

The following figure supplements are available for figure 2:

**Figure supplement 1**. Iterative HGT analysis alignments.

**Figure supplement 2**. Protein phylogeny of *A. oryzae* GH25 muramidase and relatives.

**Figure supplement 3**. DNA phylogeny of A. oryzae GH25 muramidase and relatives.

Interestingly, the lysozyme gene in the aphid *A. pisum* consists of a fusion of a bacterial GH25 muramidase domain and a eukaryotic carboxypeptidase domain. The gene includes five introns (***Nikoh et al., 2010***), none of which interrupt the GH25 domain, consistent with a relatively recent HGT event and the absence of the gene from most sequenced insects (***Figure 1***). The lysozyme in the fungus *A. oryzae*, meanwhile, contains only a single intron, but it does interrupt the GH25 domain, consistent with the domain's long association with fungi from the subkingdom Dikarya (***Figure 1***). We found no

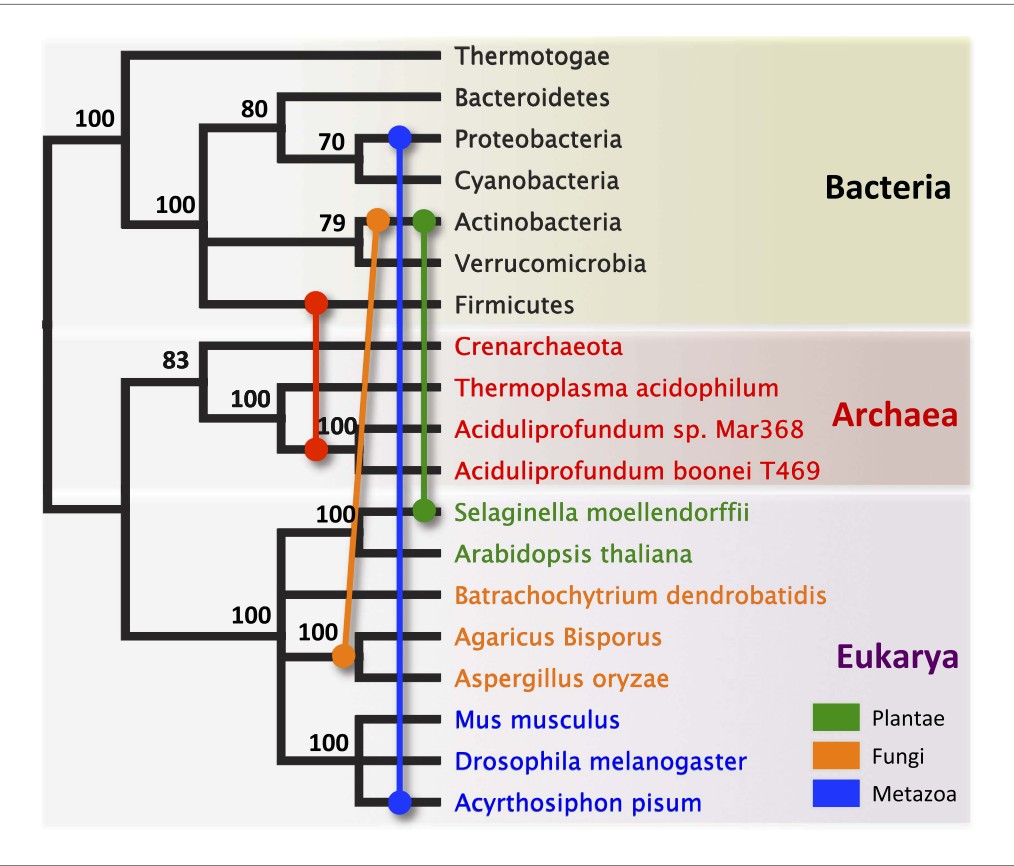

**Figure 3**. Schematic of HGT events. Bayesian phylogeny based on the 16S rRNA gene from selected taxa is shown. Colored lines indicate putative horizontal gene transfer events, although other possible HGT patterns cannot be definitively excluded. Posterior probabilities are noted at each node.

evidence of a GH25 muramidase in 323 sequenced archaeal genomes spanning all the major phyla and sister taxa to *A. boonei* (*Reysenbach et al., 2006*; *Flores et al., 2012*). This lack of homology does not appear to be due to insufficient representation of archaeal diversity, as the 323 members span all of the major phyla: Crenarchaeota (56), Euryarchaeota (205), Nanoarchaeota (10), and Thaumarchaeota (39). Indeed, if vertical descent were assumed for a recent Bayesian phylogeny of archaea with sequenced genomes (*Brochier-Armanet et al., 2011*), this would require at least 10 independent losses of the lysozyme gene, an assumption that is certainly less parsimonious than a single HGT event. Moreover, the relative divergence of the small subunit rRNA gene in *A. boonei* compared to the putative bacterial HGT donors is greater than the relative divergence of the lysozyme gene (*Figure 4*), a finding that is inconsistent with both genes being transmitted by vertical descent. Also, there are no other homologs beyond those presented in this study in 132 plant genomes, and only one insect species with additional homologs out of 109 insect genomes. Thus, if the lysozyme were present in the last common ancestor of all domains, it would require the unlikely loss of the gene in dozens of lineages while maintaining it in an exceedingly small number of species. In summary, the presence of a GH25 muramidase in nonbacterial species represents a series of recurrent, independent horizontal gene transfer events derived from diverse, ecologically associated bacteria.

## *A. boonei* GH25 muramidase is antibacterial

We next undertook a series of experiments to test the hypothesis that the transferred muramidase functions as an antibacterial molecule. Since HGT frequently results in pseudogenized and nonfunctional genes (*Kondrashov et al., 2006*; *Nikoh et al., 2010*, *2008*; *Dunning Hotopp et al., 2007*), we first investigated the amino acid sequences for preserved antibacterial action of the transferred lysozymes in nonbacterial genomes. We aligned all 86 GH25 muramidase sequences to identify

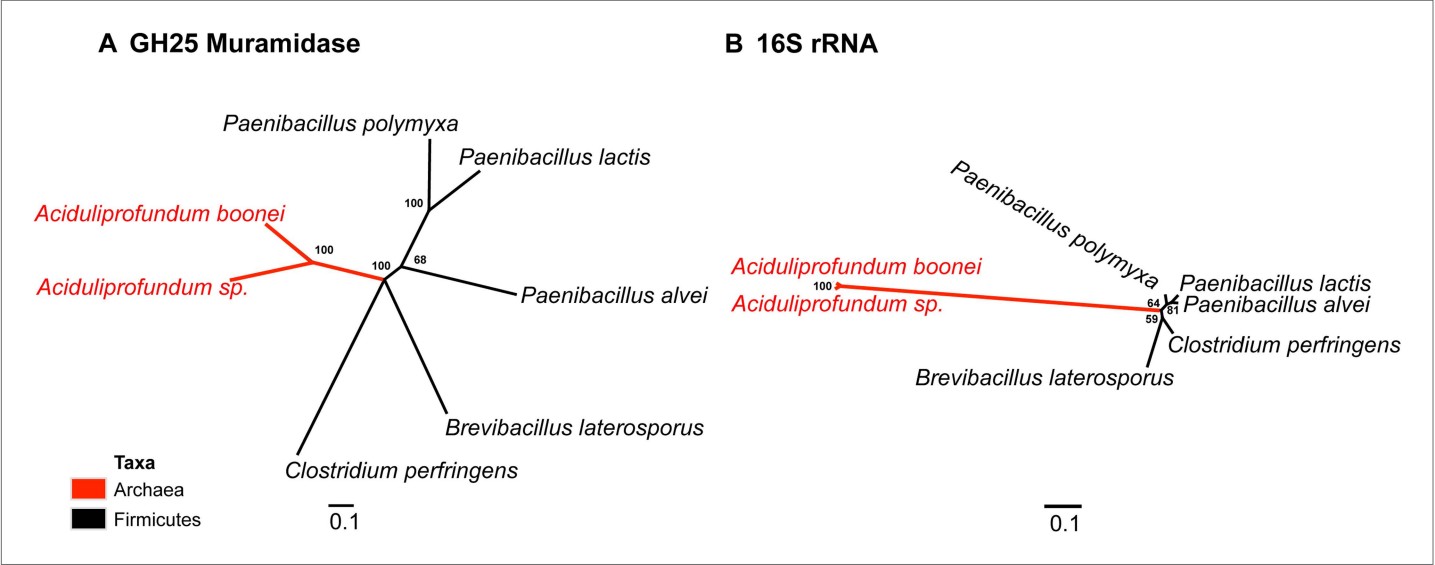

**Figure 4**. Comparison of GH25 muramidase and rRNA divergence. (**A**) Unrooted Bayesian phylogeny of the GH25 muramidase from *A. boonei* and selected relatives, based on an alignment of 185aa without indels. Taxon of origin for each nucleic acid sequence is indicated by color. Posterior probability is indicated at all nodes with values above 50. Branch lengths represent number of substitutions per site as indicated by scale bar. (**B**) Unrooted Bayesian phylogeny of the 16S rRNA gene for the same taxa as in (**A**), based on an alignment of 1,156 bp without indels.

conserved sites (*Figure 5A*). We then mapped the conserved amino acids to a three-dimensional structure prediction of the *A. boonei* GH25 muramidase domain (*Figure 5B*). Highly conserved residues (>85% identity between all taxa) invariably mapped to the previously identified active site pocket (*Martinez-Fleites et al., 2009*). Conservation was also evident for structure predictions of other GH25 muramidases in the phylogeny such as *S. moellendorffii* (*Figure 5C*).

Second, we cloned, expressed, and purified the GH25 muramidase domain from the archaeon *A. boonei* as well as from closely related homologs in *Paenibacillus polymyxa* and PhiBP. We obtained each muramidase in a pure elution (*Figure 6—figure supplement 1*) and tested for antibacterial

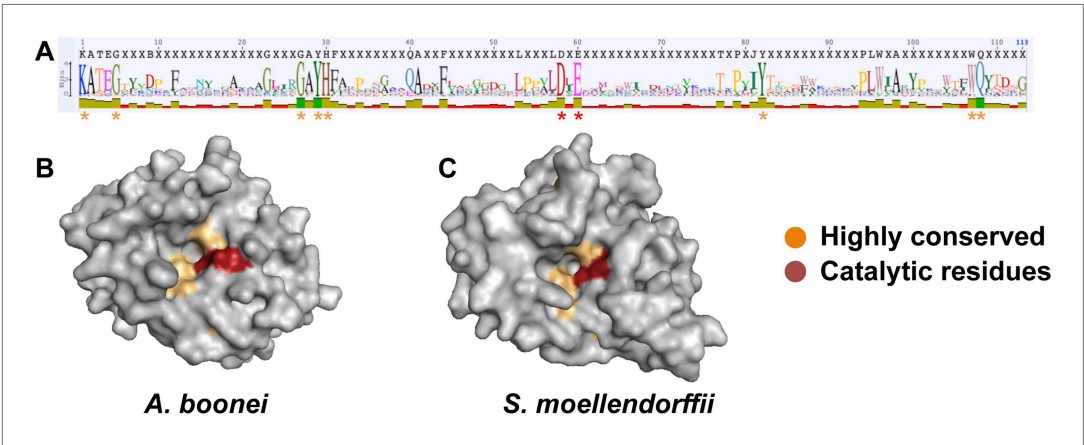

**Figure 5**. Conservation of *A. boonei* GH25 muramidase domain. (**A**) Consensus alignment of 86 GH25 muramidases with insertions and deletions removed. Conservation is indicated by amino acid symbol size and bar graphs below the consensus sequence. Active site residues and highly conserved amino acids modeled below are indicated with red and orange asterisks, respectively. (**B**) Space-filling model of the active site face of the predicted structure of *A. boonei* GH25 muramidase domain and (**C**) *S. moellendorffii* GH25 muramidase domain. Active site residues are indicated in red and the eight additional residues most highly conserved across all 86 proteins are orange.

action against a range of bacterial species. As predicted, the *A. boonei* GH25 muramidase efficiently killed several species of bacteria in the phylum Firmicutes - the putative donor group of the gene (*Figure 6A*). The bacterial inhibition by *A. boonei* GH25 muramidase was more potent than the positive control, chicken egg white lysozyme, and was dose-dependent (*Figure 6B*). Bacterial and phage muramidases did not elicit antibacterial killing, similar to cyan fluorescent protein and buffer-only negative controls. Bacteria typically use a large protein complex to limit their lysozymes' activity to the septum during cell division (*Uehara and Bernhardt, 2011*), and PhiBP phage has a documented spectrum of activity limited only to a *P. polymyxa* strain unavailable for our analyses (*Halgasova et al., 2010*). As expected, the *A. boonei* GH25 muramidase did not exhibit antibacterial activity against Gram-negative species or Gram-positive species outside of the families Bacillaceae and Paenibacillaceae, which was equivalent to the killing range of chicken egg white lysozyme with the exception of the Actinobacterium *Micrococcus luteus* (*Figure 6—figure supplement 2*).

Third, the *A. boonei* muramidase domain is part of a larger gene (1725 bp) composed of other domains that may broaden or constrain the range of antibacterial activity. To test the full-length gene's function in the absence of genetic tools in this system, we cloned the entire gene into an expression plasmid in *E. coli* and discovered that bacterial colonies grew poorly, with tiny, slow-growing colonies on solid media, and substantial cell death coinciding with a small amount of leaky expression in liquid culture (*Figure 7A*). However, two colonies grew to normal size and upon sequencing, we determined that their expression plasmids were disrupted by insertions of 774 bp (mutant 1) and 768 bp (mutant 2) of a native IS1 family transposase from *E. coli* at 21 bp or 266 bp from the start of the lysozyme gene, respectively. These insertions also resulted in a number of premature stop codons in the lysozyme reading frame, disrupting production of the full-length gene (*Figure 7B*). Thus, *E. coli* death requires intact and full-length lysozyme, and toxicity is not due to the expression construct itself. In sum, expression of the complete lysozyme resulted in *E. coli* death, while cloned genes with insertion sequences and premature stop codons abolished the lytic capacity of these proteins from within *E. coli* cells, providing evidence for an expanded host range to the antibacterial action.

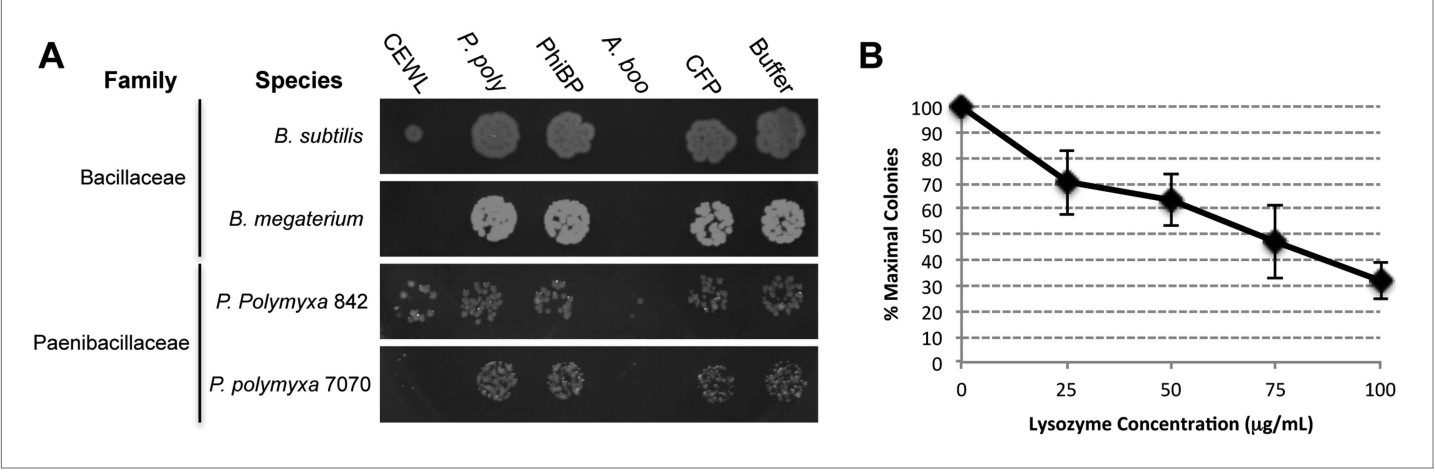

**Figure 6**. Antibacterial action of *A. boonei* GH25 muramidase domain against Firmicutes. (**A**) Bacteria of the specified strain/species incubated overnight on tryptic soy agar after a 20-min liquid preincubation with the proteins indicated. Genera: *B* = *Bacillus*, *P* = *Paenibacillus*. Proteins: CEWL = chicken egg white lysozyme, *P. poly* = *P. polymyxa* lysozyme, PhiBP = bacteriophage PhiBP lysozyme, *A. boo* = GH25 domain of *A. boonei* lysozyme, CFP = cyan fluorescent protein. Images are representative of at least three independent experiments. (**B**) Dose-dependence of *A. boonei* GH25 muramidase antibacterial action. *B. subtilis* colony survival is shown after incubation with *A. boonei* GH25 muramidase at the indicated concentrations for 20 min at 37°C. N = 10 for each concentration. p < 0.001 for linear model fit. Error bars are ± SEM.

The following figure supplements are available for figure 6:

**Figure supplement 1**. Lysozyme purifications.

**Figure supplement 2**. Antibacterial test of *A. boonei* GH25 muramidase on non-Firmicutes bacteria.

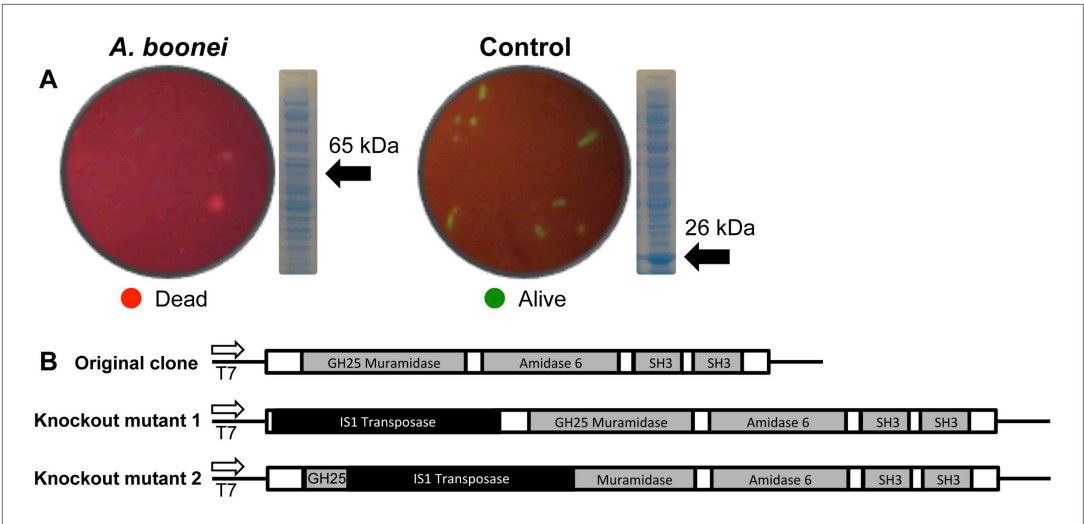

**Figure 7**. *E. coli* death following full-length *A. boonei* lysozyme expression. (**A**) Live/dead stain of BL21 (DE3) *E. coli* transformed with expression constructs for the full-length lysozyme from *A. boonei* or a control lysozyme WORiA, a bacteriophage infecting *Wolbachia pipientis* strain *w*Ri, after overnight growth without induction. PAGE gels of crude *E. coli* lysates from *E. coli* expressing the indicated lysozyme after 6 hr of induction are also shown with the expected sizes of lysozymes indicated with arrows. (**B**) Structure of original full-length *A. boonei* lysozyme expression plasmid and two spontaneous knockout mutants caused by insertion of 774 bp (mutant 1) and 768 bp (mutant 2) of IS1 transposase sequences. These insertions also resulted in a number of stop codons in the reading frame of the lysozyme. Knockout mutants grew to normal colony size, while all wild type colonies had intact expression plasmids, grew poorly, and died over time in liquid culture.

Fourth, if horizontally transferred lysozymes serve as antibacterials to fend off bacterial niche competitors, two predictions follow: the lysozyme will be upregulated in response to bacterial competition and upregulation may correlate with a relative growth advantage in coculture. We thus cultured *A. boonei* cells in anaerobic marine media (*Reysenbach et al., 2006*) with and without cohabiting *Mesoaciditoga lauensis* (phylum Thermotogae) that was isolated from the same hydrothermal vent field as *Aciduliprofundum* in the Eastern Lau Spreading Center (*Reysenbach et al., 2013*). As expected, we observed a significant increase in *A. boonei* lysozyme expression at four (up to 127% increase) and 12 hr (up to 43% increase) of coculture with *M. lauensis* in comparison to negative control cultures of the singular *A. boonei* (*Figure 8A*). Ideally, *A. boonei* wild type and lysozyme knockouts would be employed to test relative fitness and bacterial inhibition. However, genetic manipulation of *A. boonei* is not currently feasible.

Growth experiments of *A. boonei* and *M. lauensis* were continued for 72 hr, during which there was a relative Malthusian fitness (*Lenski et al., 1991*) increase for *A. boonei* in coculture vs. monoculture (*Figure 8B*) across the exponential growth phase. This difference is marginally non-significant, perhaps due to low sample sizes (p = 0.11, N = 5, MWU two-tailed test). When the species are cultured separately for 72 hr, *M. lauensis* cell abundance is greater than that of *A. boonei* during 14 out of the 19 sampling points (*Figure 8C*, blue circles), indicating that bacteria outperform archaea in monoculture conditions. However, when the two species are cocultured, the cell abundances reverse and *A. boonei* outperforms *M. lauensis* for 14 out of the 19 time points (*Figure 8D*, red circles). This competitive frequency difference is significant (Chi-square test, p = 0.0035), complementing the Malthusian fitness increase. Additionally, for each monoculture time point, there are 4.43% fewer *A. boonei* cells on average than *M. lauensis*, while in coculture there are 6.22% more *A. boonei* cells per time point (Mann Whitney U. p = 0.023). Thus, *A. boonei* outcompetes its bacterial competitor in coculture despite a higher monoculture growth rate for the bacteria, although whether lysozyme upregulation is directly responsible for this effect cannot be definitively proven. Other possible explanations for this finding include *A. boonei* scavenging of bacterial waste products, possessing a superior ability to obtain a rate-limiting nutrient, or deploying alternative antibacterial defense mechanisms.

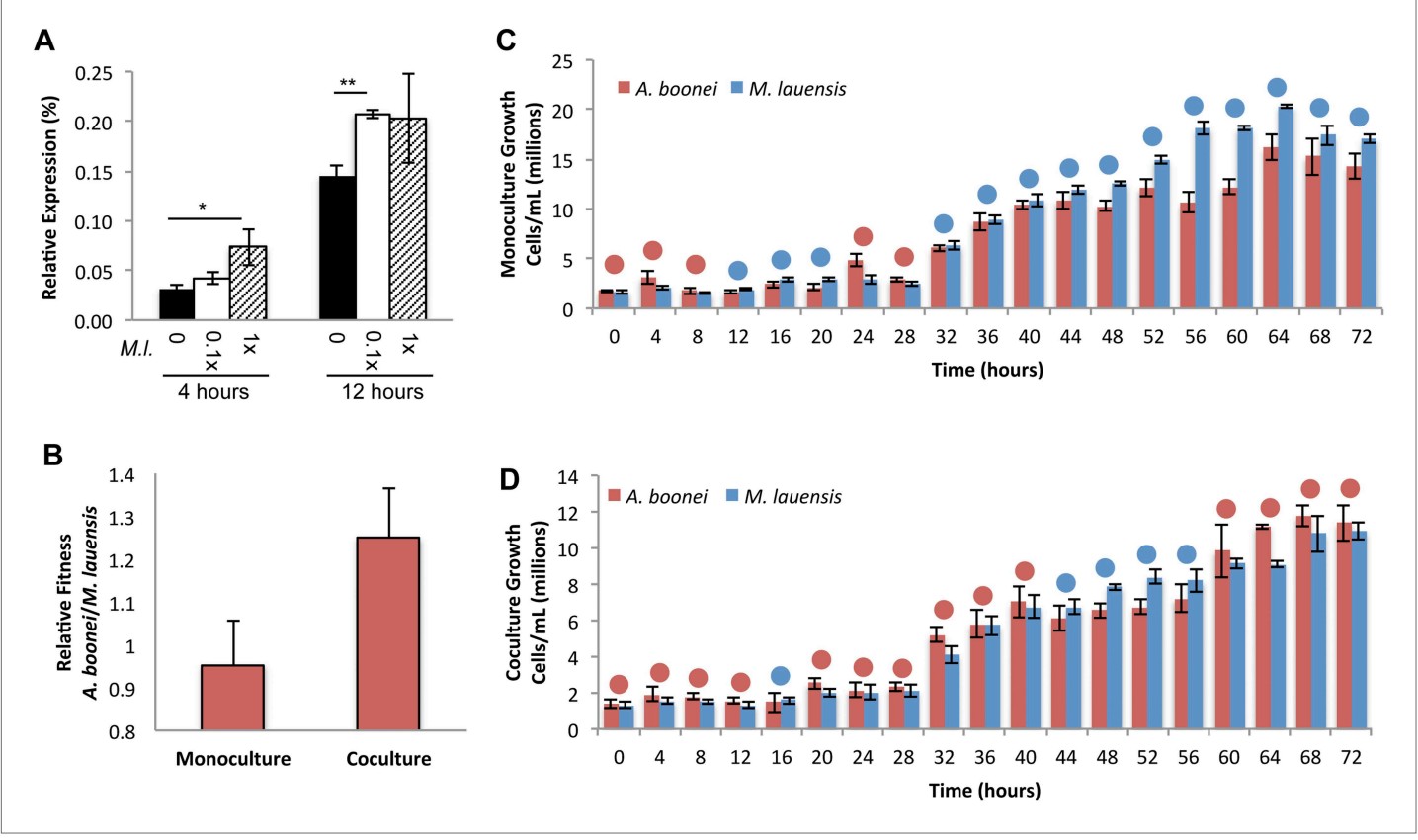

**Figure 8**. Lysozyme expression and relative fitness during *A. boonei* and *M. lauensis* coculture. (**A**) Expression of *A. boonei* GH25 muramidase relative to the control gene elongation factor 1α, after the indicated time of coculture with *M. lauensis* (M.l) at the specified ratio relative to *A. boonei*. *p < 0.05, **p < 0.01, by Mann–Whitney U pairwise comparisons. N = 6 for all samples. Primers are listed in **Supplementary file 1**. (**B**) Relative fitness of *A. boonei* vs *M. lauensis* in monoculture (N = 5) and coculture (N = 4). (**C**) Growth of *A. boonei* (red) and *M. lauensis* (blue) monocultures over time. Significant differences in cell abundance occur at 24, 52, and 64 hr (p < 0.05), and 56 and 60 hr (p < 0.01) based on pairwise Wilcoxon tests. (**D**) Growth of *A. boonei* and *M. lauensis* in coculture over time. Significant differences in cell abundance occur at 48, 52, and 64 hr (p < 0.05) based on pairwise Wilcoxon tests. Error bars are ±SEM for all panels.

## Conclusions

The universal tree represents the evolutionary relationships between cellular domains and establishes the modern foundation for benchmarking the magnitude of HGT across life. Indeed, HGTs have been described between each domain including archaea and bacteria (**Nelson et al., 1999**; **van Wolferen et al., 2013**), bacteria and eukaryote (**Andersson, 2005**; **Bordenstein, 2007**; **Gladyshev et al., 2008**; **Danchin et al., 2010**), and archaea and eukaryote (**Andersson et al., 2003**; **Schonknecht et al., 2013**). Despite these cases and others (**Brown, 2003**; **Zhaxybayeva and Doolittle, 2011**), HGTs are not without limits and often succumb to the selective costs of genomic rearrangements, cytotoxic effects, disruptive insertions, and functional inefficiencies upon integration (**Baltrus, 2013**).

It follows then that HGTs do not occur at equal rates across the universal tree, but rather experience preferential routes in which the costs of HGT are easier to overcome. The resulting pattern of HGT can be understood as a gradient of decreasing frequency from within domain > between two domains > between all domains of life (**Bruto et al., 2013**; **Zhaxybayeva and Doolittle, 2011**; **Puigbo et al., 2009**; **Andam and Gogarten, 2011**, **2013**). In support of this pattern, the overwhelming evidence of gene transfers between bacteria is counterbalanced by the extreme lack of parallel gene transfers across all extant groups of life. As these parallel transfers are usually ancient and occur in non-model organisms (**Lundin et al., 2010**; **Koonin et al., 2003**; **McClure, 2001**; **McDonald et al., 2012**; **Moran et al., 2012**), they can defy clear interpretations due to their deep antiquity and lack of functional validation.

One feature that parallel HGTs have in common is that the gene's phenotype must transcend different physiological capabilities, cellular structures, and ecological niches to repeatedly increase the fitness of each recipient across the whole diversity of life. While not traditionally used in the context of parallel HGT across all cellular domains, the term niche-transcending gene appropriately captures these conditions (*Wiedenbeck and Cohan, 2011*). The lysozyme gene family we describe in archaea, bacteria, eukaryotes, and viruses provides one such example because the adaptive benefit of an antibacterial muramidase has repeatedly surmounted the obstacles against recurrent HGT. Indeed, horizontally transferred homologs of the GH25 muramidase exhibit differential tissue expression in *A. pisum* (*Nikoh et al., 2010*) and bacteriolytic activity in the fungus *Aspergillus nidulans* (AN6470.2) (*Bauer et al., 2006*). Thus, the horizontally transferred homologs in eukaryotes confer the same transcriptional and enzymatic activity as in the archaea.

The muramidase in a thermophilic archaeon is of special note as archaea do not possess murein cell walls (*Albers and Meyer, 2011*), and genes encoding an antibacterial peptide have never before been identified (*Cantarel et al., 2009*). Members of the genus *Aciduliprofundum* are widespread thermoacidophiles in deep-sea hydrothermal vent chimney biofilms (*Flores et al., 2012*) in which bacteria are frequent inhabitants (*Orcutt et al., 2011a*; *Miroshnichenko and Bonch-Osmolovskaya 2006*), including the *M. lauensis* species tested above. Archaea have been largely ignored in the context of antibiotic discovery, likely because of the conjecture that archaea do not compete with bacteria in nature. However, given that they coexist with diverse bacterial species in the environment (*Oren, 2002*; *Kato and Watanabe, 2010*; *Orcutt et al., 2011b*) and can compete for similar resources, there may be significant, unexploited potential for antibiotics in this domain. Only a handful of antimicrobial peptides produced by archaea have been characterized, and those are active only against other archaea (*O'Connor and Shand 2002*) despite the fact that archaea are known to inhibit bacteria in diverse environments (*Atanasova et al., 2013*; *Shand and Levya, 2008*). It is also possible that since *Aciduliprofundum* strains metabolize peptides, the lysozyme enables a nutritive strategy in which lysed bacteria provide nutrients for the archaeon to scavenge.

Based on this work, we suspect that systematic surveys of archaeal gene products will likely uncover a broad range of antibacterial activities, and may eventually offer novel peptide or small molecule therapeutics. Such antibacterial products may have naturally evolved thermostability that would increase their attractiveness as therapeutics. GH25 muramidases have been demonstrated as effective antibacterials against biofilms of *Streptococcus pneumoniae* (*Domenech et al., 2011*) and related enzymes have proven efficacious in mouse models of bacterial mucosal colonization (*Fenton et al., 2010*), sepsis (*Loeffler et al., 2003*), and endocarditis (*Entenza et al., 2005*).

In summary, we infer that the evolutionary path to this parallel HGT was paved by the universal drive for nonbacterial taxa to compete in a bacterial world. We predict that similar to the cascade of antibiotic gene transfer discoveries that followed their initial reporting, parallel transfers of genes to all cellular domains and viruses might regularly have antimicrobial functions.

## Materials and methods

Unless otherwise stated, reagents were obtained from Fisher Scientific (Waltham, WI).

### PCR and sequencing

PCR was performed using GoTaq DNA Polymerase (Promega, Madison, WI) with primers listed in *Supplementary file 1*. PCR products were electrophoresed using 1% agarose gels in sodium boric acid buffer. Following electrophoresis, gels were dyed with GelRed (Phenix Research, Candler, NC) and imaged on an Alpha Innotech GelRed Imager (Alpha Innotech, San Leandro, CA). Amplified bands were excised from the gels and purified with an SV Wizard Gel Cleanup kit (Promega). Following purification, DNA concentration was measured using the Qubit DNA high sensitivity kit (Life Technologies, Grand Island, NY) and sequencing reactions were performed by Genewiz (South Plainfield, NJ).

### Bioinformatics

The lysozyme protein from *Wolbachia* prophage WORiA (ZP_00372884) was used as a query in a blastp search of the NCBI nonredundant protein database using Geneious Pro v5.5.6. All hits with E-values below $10^{-12}$ were collected and duplicate entries were removed. Sequences from field and laboratory samples were added to this collection and aligned with MUSCLE (*Edgar, 2004*), insertions and deletions were removed, and the eight most highly conserved residues from the MUSCLE

alignment were mapped to a structure prediction of *A. boonei* lysozyme using PyMOL. Structure prediction was performed using the homology-based modeling tool Phyre2 (*Kelley and Sternberg, 2009*). For phylogenetic analyses, ProtTest (*Abascal et al., 2005*) was used to determine the best model of protein evolution based on the corrected Akaike information criterion (AICc). MrBayes (*Ronquist et al., 2012*) and PhyML (*Guindon et al., 2010*) were used to build a phylogenetic tree with Bayesian and maximum likelihood methods, respectively. For the global lysozyme phylogeny, the best model chosen by ProtTest (LG + I + G) was used to generate the maximum likelihood tree, while the third best model (WAG + I + G; ΔAICc: 74.82) was used to generate the Bayesian tree due to a lack of LG model availability in MrBayes. *S. sanguinolenta* and *S. stauntoniana* lysozymes were excluded from this analysis because frameshift mutations suggest the genes may be evolving in the absence of selection, while Aphidinae lysozymes were not included because of shorter sequences of the GH25 muramidase domain obtained through the use of degenerate primers that would have limited resolution of the tree.

In an iterative approach, each candidate example of HGT was used as a blastp query against the nr database and the top 15 (*A. boonei*, *A. pisum*, *S. moellendorffii*) or top 75 (*A. oryzae*) E-value hits were subjected to the same phylogenetic analysis as above. Evolutionary models used were *A. boonei*: WAG + I + G on a 148aa indel-free alignment, *A. pisum*: CpREV + I + G on a 190aa indel-free alignment, *S. moellendorffii*: WAG + I + G (ΔAICc: 44.28) on a 200aa indel-free alignment, *A. oryzae*: WAG + G (Bayesian, ΔAICc: 4.49) or LG + G (maximum likelihood) on a 186aa indel-free alignment. The fungal lysozyme was also phylogenetically analyzed on the DNA level using the top 25 E-value blastn hits to exon 2 of the *A. oryzae* lysozyme gene. jModelTest 2 (*Darriba et al., 2012*) was used to determine the best model of nucleic acid evolution (GTR + I + G, ΔAICc: 7.33) of a 282 bp indel-free alignment. The archaea HGT clade was also analyzed phylogenetically with a Bayesian tree of selected taxa using lysozyme protein sequences (WAG + I + G, 185aa) and compared to 16S rRNA (GTR + G, 1,156 bp) for the same strains obtained from SILVA (*Quast et al., 2013*). A schematic representation of putative HGT events was plotted on a Bayesian phylogeny based on the 16 s rRNA gene (GTR + G, 1,226 bp). Representative species used for this phylogeny were *Magnetospirillum magneticum* (Proteobacteria), *Paenibacillus polymyxa* (Firmicutes), *Arthrospira maxima* (Cyanobacteria), Crenarchaeota archaeon SCGC AAA471-B05, *Chthoniobacter flavus* (Verrucomicrobia), *Thermotoga maritima* (Thermotogae), *Pedobacter saltans* (Bacteroidetes), *Streptomyces violaceusniger* (Actinobacteria).

Statistical support for the HGT hypothesis was assessed with the Shimodaira–Hasegawa test (SH-test) (*Shimodaira and Hasegawa, 1999*) as implemented in RAxML v.8.0.20 (*Stamatakis, 2014*). An unresolved binary constraint tree was generated in MacClade v4.08, in which bacterial sequences are monophyletic, as are nonbacterial sequences, with all other topology unconstrained. This constraint tree was used to generate a maximum likelihood best tree with RAxML, using the same evolutionary models as above. The SH-test was then run comparing the maximum likelihood constrained tree to the unconstrained consensus Bayesian tree or to 100 bootstrap trees from the maximum likelihood analysis from PhyML. This procedure was repeated for a three-domain constraint tree consistent with the tree of life and with a constraint tree in which eukaryotic sequences were monophyletic and other sequences were unconstrained.

## Lysozyme cloning and purification

*A. boonei* GH25 muramidase domain (ZP_04874596), *P. polymyxa* lysozyme (YP_003869492), and PhiBP lysozyme (CBA18122) were cloned and expressed with a 6x C-terminal histidine tag using an Expresso T7 Cloning and Expression System (Lucigen, Middleton, WI) according to the manufacturer's instructions. We also cloned the *S. moellendorffii* and *A. oryzae* GH25 muramidases, however recombinant 6× histidine-tagged proteins were insoluble when expressed in either *E. coli* or sf9 insect cells and attempts to solubilize them were unsuccessful. Sequence-confirmed expression plasmids and a control plasmid expressing cyan fluorescent protein (CFP) were transformed into HI-Control BL21 (DE3) *E. coli* cells. Cultures at an OD600 of ~0.5 were induced with 1 mM IPTG for 6 hr, centrifuged, and frozen at −80°C until purification. Frozen pellets were resuspended in lysis buffer containing 10 mM Tris–HCl, pH 7.5, 300 mM NaCl, 0.5% Triton X-100, 0.3% sodium dodecyl sulfate, and 1 mM phenylmethylsulfonylfluoride and sonicated 5 times for 30 s with at least 1 min on ice between sonications. Samples were centrifuged and recombinant proteins were purified from supernatant using HisPur Ni-NTA chromatography cartridges (Thermo Scientific, Waltham, MA) according to manufacturer's instructions. Glycerol at a final concentration of 40% was added to enzymes in elution buffer for

storage at −20°C for a maximum of 3 weeks before use in antibacterial assays. Purifications were analyzed with denaturing polyacrylamide gel electrophoresis and stained with GelCode Blue (Thermo Scientific).

Full-length *A. boonei* lysozyme and WORiA lysozyme were cloned into a pET-20b vector (EMD Millipore, Darmstadt, Germany) with a C-terminal 6× histidine tag and sequence-confirmed plasmids were transformed into BL21 (DE3) *E. coli* (EMD Millipore). Three colonies from each transformation were inoculated into *LB* media and grown to an OD600 of ~0.5, induced for 4 hr with 1 mM IPTG and harvested for analysis on PAGE gels. Overnight cultures without induction were examined for bacterial death with a BacLight Live/Dead Stain (Life Technologies).

## Antibacterial assays

Purified *A. boonei* GH25 muramidase, *P. polymyxa* lysozyme, PhiBP lysozyme, CFP, and commercially purchased CEWL (Sigma-Aldrich, St. Louis, MO) were diluted to 100 µg/ml in buffer EG (60% nickel column elution buffer, 40% glycerol) and filter sterilized. Bacteria to be tested were grown overnight in tryptic soy broth, split 1:10, and incubated to exponential growth before being diluted into each enzyme solution. Samples were incubated with shaking for 20 min at 37°C and then 5 µl was spotted onto tryptic soy agar and incubated overnight at 37°C. To evaluate whether antibacterial activity is dose-dependent, *B. subtilis* was incubated with *A. boonei* GH25 muramidase at 100 µg/ml, 75 µg/ml, 50 µg/ml, 25 µg/ml and 0 µg/ml and 100 µl was spread on tryptic soy agar plates. Replicates of 10 were performed for each concentration, plates were incubated overnight at 37°C, and colonies were counted the following morning. Bacterial strains used in these experiments are listed in *Supplementary file 1*.

## A. boonei cultures

*A. boonei* and *M. lauensis* cultures were performed as previously described (*Reysenbach et al. 2006*) with the following modifications: yeast extract was added at 2.0 g/l, pH was adjusted to 4.8, and cultures were incubated at 65°C. For gene expression studies, $8.2 \times 10^5$ cells were inoculated into 5 ml cultures in 6 replicates each of monocultures and cocultures at 0.1:1, 1:1, and 1:0.1 ratios and 500 µl samples were collected after 4 and 12 hr of co-incubation and frozen for expression analysis. RNA was isolated from frozen samples using an RNeasy Mini Kit (Qiagen) and QIAshredder (Qiagen), DNA contamination was removed with a Turbo DNAfree Kit (Life Technologies), and reverse transcription was performed using a Superscript III first Strand Synthesis System (Life Technologies) along with no-reverse transcriptase controls. Quantitative PCR was performed with GoTaq qPCR Master Mix (Promega) using a CFX96 Real-Time System (Bio-Rad, Hercules, CA). Primers are listed in *Supplementary file 1*. For competition studies, 5 replicates of 5 ml cultures were inoculated as monocultures or 1:1 cocultures and 175 µl was collected every 4 hr for counting of relative species abundance with a hemocytometer. Relative fitness was calculated based on Malthusian parameters over the period of exponential growth as previously described (*Lenski et al., 1991*).

## Acknowledgements

We are grateful to members of the Bordenstein and Reysenbach labs for helpful discussions, and particularly to Christine Sislak for *Mesoaciditoga lauensis* cultures, Sarah Bordenstein for WO sequences, and Jessica Hardwicke for assistance with cell counting. We also thank Antonis Rokas, Jennifer Wisecleaver, Kristin Jernigan, and Elise Pfaltzgraff for technical assistance, and Julie Dunning Hotopp and Barton Slatko for helpful feedback on an earlier version of this manuscript, as well as Patrick Abbot for aphid samples and manuscript feedback, and Robert Brucker for support with figures.

## Additional information

### Funding

| Funder | Grant reference number | Author |
| --- | --- | --- |
| National Science Foundation | DEB1046149 | Seth R Bordenstein |
| National Institutes of Health | R01GM085163 | Seth R Bordenstein |

| Funder | Grant reference number | Author |
|---|---|---|
| National Institutes of Health | T32GM07347 | Jason A Metcalf |
| National Science Foundation | DEB1134877 | Anna-Louise Reysenbach |

The funders had no role in study design, data collection and interpretation, or the decision to submit the work for publication.

### Author contributions

JAM, Conception and design, Acquisition of data, Analysis and interpretation of data, Drafting or revising the article; LJF-J, KB, Acquisition of data, Analysis and interpretation of data, Drafting or revising the article; A-LR, SRB, Conception and design, Analysis and interpretation of data, Drafting or revising the article, Contributed unpublished essential data or reagents

## Additional files

### Supplementary files

• Supplementary file 1. Primers, field samples, and bacterial strains used in this study.

• Supplementary file 2. Identity matrix of GH25 muramidases.

• Supplementary file 3. Bordering gene BLAST results.

• Supplementary file 4. Iterative HGT analysis BLAST results.

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
