## [Decision Letter]

Thank you for sending your work entitled “Antibacterial Gene Transfer Across the Tree of Life” for consideration at *eLife*. Your article has been favorably evaluated by Diethard Tautz (Senior editor), Wilfred van der Donk (Reviewing editor), and 3 reviewers (Peter Gogarten, Anthony Poole, and David Baltrus). The Reviewing editor and the reviewers discussed their comments before we reached this decision, and the Reviewing editor has assembled the following comments to help you prepare a revised submission.

The study provides clear evidence for the transfer of the GH25 muramidase between the three domains of life. Phylogenetic analyses establish that the eukaryotic and archaeal versions of the gene have bacterial neighbors and that the transfers occurred more recently than the splitting of the domains. Furthermore, the use of phenotypic tests as well as expression in *E. coli* establishes that the GH25 muramidase is functional across its range and not a pseudogene or other non-functional relic. The assessment of genomic context also supports that the GH25 muramidase gene has been transferred from bacteria to archaeal and eukaryote recipients.

The potential antibacterial role of the GH25 muramidase gene in the archaeal recipient Aciduliprofundum boonei is addressed by showing that the product of the transferred gene exhibits antibiotic properties against expected species in purified form. Experimental work in a non-model system that lacks standard genetic tools is challenging, so these experiments along with the data in *E. coli* provide a good means of assessing function.

Finally, the authors test whether the gene can impact competitors. Again, without the capacity to undertake direct knockouts, the authors do an admirable job of testing this, by examining relative growth in both monocultures and cocultures with Mesoaciditoga lauensis, a bacterium known to cohabit the environments from which A. boonei has been isolated.

All reviewers agreed that this study is of high interest to the readership of *eLife*, but request you address the following points in a revised manuscript:

1) One question stems from the observation that not all field isolates appear to have lysozyme genes. If there are isolates that genuinely lack the GH25 muramidase genes, these would make a good control. It would be helpful if the authors could comment on whether they think the isolates definitely lack the GH25 muramidase gene, and whether they have tried the growth and coculture assays with any of these isolates.

2) The clarity of the manuscript would benefit greatly from the inclusion of a single phylogeny (recA-like genes?) across all domains of life with arrows actually showing putative transfer directions. The included phylogenies are great, but don't necessarily give an intuitive sense of how distant these transfers are. The inclusion of non-lysozyme containing taxa would really help tell the story to a wider audience.

3) The SH tests of phylogenies are significantly different and demonstrate that the constrained tree is a poorer fit than the unconstrained tree. However, this only demonstrates that at least one transfer has occurred while the study is about numerous independent transfers...it is possible to sequentially constrain the tree and perform independent SH tests for each potential transfer, which would make a stronger story.

4) In regards to the competition assays, the reviewers realize that the authors are dealing with intractable genetic systems, but were wondering if there might be cleaner ways of comparing growth between microbes, instead of independent vs. together. The current comparison is confounded because a variety of lysozyme independent reasons could be responsible for a shift in competitive outcome in co-culture (i.e. cross-feeding of archaea on bacterial waste products). Would it be possible to isolate lysozyme resistant bacteria and perform paired competition assays with sensitive and resistant versions? The reviewers felt that the story might be good enough as is without such experiments but wanted to share their thoughts with the authors. Regardless, the authors should point out the caveats of their approach in the paper.

---

## [Author Response]

*1) One question stems from the observation that not all field isolates appear to have lysozyme genes. If there are isolates that genuinely lack the GH25 muramidase genes, these would make a good control. It would be helpful if the authors could comment on whether they think the isolates definitely lack the GH25 muramidase gene, and whether they have tried the growth and coculture assays with any of these isolates*.

The idea of using an *Aciduliprofundum* strain without GH25 muramidase in coculture experiments is an excellent one. Unfortunately, it is not clear that the strains that lacked GH25 muramidase by PCR truly do not possess the gene. Two species of *Aciduliprofundum* have been fully sequenced, T469 from the Lau Spreading Center in the Western Pacific Ocean and Mar08-339 from the Mid-Atlantic Ridge. The lysozyme sequences from these two strains have only 58% pairwise identity at the nucleotide level, suggesting that there has been substantial divergence between geographically disparate species. Thus, although PCR using degenerate primers designed for both sequenced species was able to detect close relatives in the same vent fields, we suspect that the PCR-negative *Aciduliprofundum* strains likely also have the lysozyme. We have updated the text to reflect this possibility.

*2) The clarity of the manuscript would benefit greatly from the inclusion of a single phylogeny (recA-like genes?) across all domains of life with arrows actually showing putative transfer directions. The included phylogenies are great, but don't necessarily give an intuitive sense of how distant these transfers are. The inclusion of non-lysozyme containing taxa would really help tell the story to a wider audience*.

We thank the reviewer for the suggestion and agree that this phylogeny would be a good way to illustrate the HGT events described in the manuscript. We have added it as Figure 3.

*3) The SH tests of phylogenies are significantly different and demonstrate that the constrained tree is a poorer fit than the unconstrained tree. However, this only demonstrates that at least one transfer has occurred while the study is about numerous independent transfers...it is possible to sequentially constrain the tree and perform independent SH tests for each potential transfer, which would make a stronger story*.

The reviewers are correct that additional SH tests would strengthen the statistical support for multiple HGT events. We have now performed two more SH tests between the lysozyme HGT tree and either 1) a three-domain constraint tree, or 2) a constraint tree in which the eukaryote branch is monophyletic and other branches are unconstrained. In both cases, the constrained trees were statistically inferior to the HGT trees at P < 0.01. SH tests using monophyletic constraints on individual HGT clades (i.e., plant and bacteria) are not suitable in these individual cases because the transferred genes are monophyletic in both the constraint tree and the HGT tree.

However, the recurrent patchy distribution of these lysozyme genes and the comparison between branch lengths in lysozyme phylogenies compared to 16S phylogenies further support the hypothesis of HGT, as discussed in the manuscript.

*4) In regards to the competition assays, the reviewers realize that the authors are dealing with intractable genetic systems, but were wondering if there might be cleaner ways of comparing growth between microbes, instead of independent vs. together. The current comparison is confounded because a variety of lysozyme independent reasons could be responsible for a shift in competitive outcome in co-culture (i.e. cross-feeding of archaea on bacterial waste products). Would it be possible to isolate lysozyme resistant bacteria and perform paired competition assays with sensitive and resistant versions? The reviewers felt that the story might be good enough as is without such experiments but wanted to share their thoughts with the authors. Regardless, the authors should point out the caveats of their approach in the paper*.

We thank the reviewers for this perceptive analysis. We agree that there may be reasons other than lysozyme-dependent inhibition of bacterial competitors for the results seen in our coculture assays. We have added these caveats to the manuscript in addition to the original ones already stated. While the experiments suggested are interesting, the poor availability of cultured microbes that can subsist in these extreme culture conditions, as well as the need to test such strains for lysozyme susceptibility, make such studies outside the scope of the current work.